# Water-Soluble Single-Benzene Chromophores: Excited State Dynamics and Fluorescence Detection

**DOI:** 10.3390/molecules27175522

**Published:** 2022-08-27

**Authors:** Yingge Fan, Jin Ma, Huijing Liu, Taihong Liu

**Affiliations:** 1Department of Chemical Engineering, Shaanxi Institute of Technology, Xi’an 710300, China; 2Key Laboratory of Applied Surface and Colloid Chemistry of Ministry of Education, School of Chemistry and Chemical Engineering, Shaanxi Normal University, Xi’an 710062, China

**Keywords:** single-benzene chromophore, femtosecond transient absorption, excited state dynamic, fluorescence detection, visual identification

## Abstract

Two water-soluble single-benzene-based chromophores, 2,5-di(azetidine-1-yl)-tereph- thalic acid (**DAPA**) and its disodium carboxylate (**DAP-Na**), were conveniently obtained. Both chromophores preserved moderate quantum yields in a wide range of polar and protonic solvents. Spectroscopic studies demonstrated that **DAPA** exhibited red luminescence as well as large Stokes shift (>200 nm) in aqueous solutions. Femtosecond transient absorption spectra illustrated quadrupolar **DAPA** usually involved the formation of an intramolecular charge transfer state. Its Frank–Condon state could be rapidly relaxed to a slight symmetry-breaking state upon light excitation following the solvent relaxation, then the slight charge separation may occur and the charge localization became partially asymmetrical in polar environments. Density functional theory (DFT) calculation results were supported well with the experimental measurements. Unique pH-dependent fluorescent properties endows the two chromophores with rapid, highly selective, and sensitive responses to the amino acids in aqueous media. In detail, **DAPA** served as a fluorescence turn-on probe with a detection limit (DL) of 0.50 μM for Arg and with that of 0.41 μM for Lys. In contrast, **DAP-Na** featured bright green luminescence and showed fluorescence turn-off responses to Asp and Glu with the DLs of 0.12 μM and 0.16 μM, respectively. Meanwhile, these two simple-structure probes exhibited strong anti-interference ability towards other natural amino acids and realized visual identification of specific analytes. The present work helps to understand the photophysic–structure relationship of these kinds of compounds and render their fluorescent detection applications.

## 1. Introduction

Amino acids are the fundamental building blocks of biological macromolecular proteins and play pivotal roles in many physiological processes and behaviors [1,2,3]. According to the structural characteristics of side groups, 20 common natural amino acids generally fall into four categories, i.e., hydrophobic, polar, and acidic as well as basic ones. Aspartic acid (Asp) and glutamic acid (Glu) are typical acidic amino acids, and are involved in many physiological processes, such as learning, memory, movement disorders, and other brain functions [4,5,6]. Arginine (Arg), lysine (Lys), and histidine (His) are well-known basic amino acids and play crucial roles in many biological processes such as cell division, healing of wounds, release of hormones, the immune system, and metabolism, etc. [7,8,9]. For various amino acids, maintaining an appropriate level in biological systems is of great importance and any serious alterations may cause related diseases. For example, high concentrations of Asp may cause motor neuron disease known as Lou-Gehrig’ s disease [10], while excessive Lys in urine and plasma could even lead to cystinuria or hyperlysinemia [11,12]. Therefore, the development of versatile sensor systems for discriminating and quantifying of various amino acids becomes more important for human health and medical diagnosis of diseases.

To date, a variety of sensing technologies have been developed to detect amino acids, including high-performance liquid/ion chromatography [13,14], amperometric enzyme electrodes [15,16], and diverse spectroscopic approaches, etc. [17,18,19]. Among the current detection methods, fluorescence methods have drawn increasing attention owing to their simple operation system, inherent sensitivity, fast response, and non-invasiveness in bio-samples [20,21,22,23]. A variety of fluorescent sensors such as metal-containing complexes [24,25], quantum dots [26,27], metal nanoparticles [28], π-conjugated organic dyes [29,30,31,32,33], and surfactant-assisted assemblies [34] have been intensively investigated to discriminate amino acids. In particular, shifting the emission wavelength towards the far-red and even the near-infrared (NIR) region is a requisite for deeper in-vivo biological imaging. Advantages include strong absorption and high photo-stability, combined with the infinite possibilities offered by organic synthesis to functionalize the single-benzene chromophore and to tune their spectroscopic properties. For example, Zhang et al. developed a new 1,8-naphthalimide-Cu(II) ensemble and utilized it for sensitive detection of thiols-containing cysteine (Cys), histidine (His), and glutathione (GSH) at pH 7.4 in organic media [35]. Tuccitto et al. reported a new class of fluorescent carbon quantum dots (CDs) for sensing hydrophobic amino acids from polar ones based on the interaction between the surface of activated CDs and analytes [36]. Chen and co-workers designed and synthesized a novel TCF-imidazo [1,5-α]pyridine-based fluorescent probe, which exhibited high sensitivity and excellent selectivity toward GSH over other related bio-species including Cys and homocysteine (Hcy) [37]. Moreover, we previously reported two unique ternary sensor systems based on fluorophore-anionic surfactant-Cu^2+^ assemblies, allowing rapid and selective discrimination of basic amino acids such as Arg and Lys in aqueous solution [38,39]. However, most of these fluorescent probes targeted thiol-containing amino acids, and their molecular structures are somewhat complicated and required laborious chemical synthesis procedures. Therefore, the development of new fluorescent probes with compact structures that are sensitive and selective to monitor various amino acids remains meaningful.

Featuring distinct emissive properties as well as simple molecular structures, the chromophores with single-benzene-based skeletons are potentially applied in optoelectronic devices and fluorescence sensing [40,41,42,43,44,45,46,47]. Our group firstly reported the diethyl 2,5-di(azetidine-1-yl)terephthalate derivatives as representative single-benzene chromophores in 2019 [44]. With small π-conjugated systems, these compounds impressively emit intense luminescence in both solution and solid states. This observation further confirmed that introduction of four-membered azetidine as an electron-donating group leads to effectively extended π-conjugation and enhanced brightness of a variety of chromophores [48,49]. Bondar et al. recently developed a new squaraine derivative, 2,4-bis[4 -(azetidyl)-2-hydroxyphenyl]squaraine, which displayed efficient NIR emission, large two-photon absorption (2PA) cross sections, and high photostability [50]. Aiming to obtain water-soluble and environment-sensitive fluorescent probes, we developed two new single-benzene derivatives, 2,5-di(azetidine-1-yl)terephthalic acid (**DAPA**) and its disodium carboxylate (**DAP-Na**) (Figure 1). Besides compact molecular structure and easy synthesis, both chromophores demonstrated efficient luminescence and large Stokes shifts in most highly polar and protonic solvents. More importantly, the superior pH-responsive fluorescence emission of **DAPA** and **DAP-Na** can be successfully used for efficient identification of different amino acids in aqueous media.

## 2. Results and Discussion

### 2.1. Photophysical Properties of ***DAPA*** and ***DAP-Na***

UV−vis absorption and fluorescence emission spectra of water-soluble **DAPA** and **DAP-Na** were firstly investigated in water and the results are shown in Figure 1a. The absorption maximum of **DAPA** was located at 365 nm, while that of **DAP-Na** was blue-shifted and centered at 338 nm. The emission peak of **DAPA** was located at 591 nm and exhibited an obviously bathochromic shift with respect to that of **DAP-Na** (535 nm). The obtained bathochromic shift phenomena are probably due to the stronger intramolecular charge transfer property [49]. Typically, these two single-benzene-based chromophores feature extraordinarily large Stokes shifts of up to 226 nm in water. This special characteristic and almost zero overlap between their excitation and emission spectra (Appendix A) commonly benefit fluorescent sensing applications. Of particular interest, **DAP-Na** displayed intensively green fluorescence with a quantum yield (Φ_f_) of up to 0.42 in aqueous media. However, **DAPA** exhibited relatively weak red emission with Φ_f_ around 0.08.

Photophysical properties of these two single-benzene-based chromophores were investigated in highly polar and protonic solvents including DMF, DMSO, EtOH, and MeOH. **DAPA** showed obviously solvent-dependent properties and was characterized by single absorption maxima around 457 nm in DMF and 470 nm in DMSO, respectively (Figure 1b). In EtOH and MeOH, its absorption spectra at longer wavelengths were characterized by two bands located at 415–550 and 305–415 nm. Based on our understanding, the dicarboxylic acid of **DAPA** may be partially ionized, resulting in two balanced components (ionized state and carboxylic acid state) in the abovementioned solutions. If further considering the polarity of MeOH is larger than that of EtOH, the proportion of the component of ionized state (377 nm) in MeOH was slightly higher than that of carboxylic acid state (456 nm) as shown in Figure 1b. However, in water, the carboxy on one side was completely ionized, and thus **DAPA** was characterized by single absorption maxima around 365 nm. However, the absorption maxima of **DAP-Na** were insensitive to solvent properties (Appendix A), and fluorescence emission of **DAPA** and **DAP-Na** exhibited comparatively slight dependence on the abovementioned solvents (Figure 1c and Appendix A). Importantly, **DAPA** and **DAP-Na** demonstrated moderate Φ_f_ values within a range of 0.20–0.61 in all the tested organic solvents (Table 1).

### 2.2. Theoretical Calculations

Time-dependent DFT calculations of **DAPA** and **DAP-Na** were performed in water at the CAM-B3LYP/6-31G(d,p) level to investigate their structural characteristics and photophysical properties (Figure 2a). The molecular orbital diagrams of **DAPA** and **DPA-Na** in both ground and excited states demonstrated HOMOs that were mainly localized over the benzene ring and the azetidines. However, LUMOs moved to the electron-withdrawing carbonyl moiety, indicating an effective push–pull system was established for these two single-benzene frameworks (Figure 2b,c). It is noteworthy that the absorptive energy gap of **DAP-Na** (6.48 eV) between the ground HOMO (−5.90 eV) and LUMO (0.58 eV) was larger than that of **DAPA** (5.94 eV). The similar trend of emissive energy gaps explicated the bathochromic shift phenomena from **DAP-Na** to **DAPA**. Moreover, their theoretical maxima absorption and emission bands agreed well with the experimental results, supporting our understanding about their photophysics.

### 2.3. Transient Absorption Spectroscopy

To obtain a full understanding of the excited state dynamics, femtosecond transient absorption (fs-TA) measurements of **DAPA** were carried out in DMF, DMSO, EtOH, and MeOH upon an excitation at fs-420 nm. fs-TA spectra recorded at less than 0.5 ps delay time were characterized by a positive excited state absorption (ESA) band in the 470–850 nm region and accompanied by three peaks at 495, 610, and 790 nm (Figure 3a,c and Appendix A). As mentioned in previous reports, highly symmetric push–pull chromophores, such as **DAPA**, could be recognized as quadrupolar molecules and the solvent dependence of their fluorescence is usually related to the formation of an intramolecular charge transfer (ICT) state [51,52,53,54,55]. In our opinion, the Frank–Condon S1 state of **DAPA** could be rapidly relaxed to a slight symmetry-breaking state upon light excitation following the solvent relaxation, and then the slight charge separation may occur and the charge localization become partially asymmetrical in polar media [56,57]. To provide an overview of excited-state dynamics and charge transfer properties of the fluorophores, the decay curves in fs-TA spectra and multi-exponential fitting kinetic traces probed at different wavelengths were compared as shown in Figure 3b and Appendix A. The corresponding lifetimes of transient states are summarized in Appendix A. Based on these spectral traces and time constants, a corresponding sequential model for global fitting was proposed as shown in Figure 3d. The first decay lifetime may reflect the time-constant of solvent relaxation (τ*_SR_* < 1 ps). The second one is associated with the formation of the charge separation state and symmetry-breaking state (τ*_CS_*_~_5 ps), implying the ICT localizing on the partial donor (azetdiine)-π-acceptor (carboxy) branch. The falling processes reflected the formation of the charge recombination process and the quenching of the excited state absorption (ESA) process (Appendix A). Comparatively, the fs-TA spectra of parent compound **5** (diethyl 2,5-di(azetidine-1-yl)terephthalate) in DMSO displayed similarly spectral characteristics and photoinduced excited-state dynamics (Appendix A). These results remarkably revealed that the ICT state can also be formed for these symmetric push–pull single-benzene chromophores based on the view of excited-state dynamics.

### 2.4. pH-Dependent Fluorescence

As shown in Appendix A, in the NMR spectrum for **DAPA**, there were two split peaks at 3.72 and 3.25 ppm assigned to the proton (-NC*H*_2_-) close to the N atom. Based on our understanding, this special splitting should be originated from the formed hydrogen bonding between the carboxyl-H and N atom, thus resulting in the asymmetry or twist of the azetidine ring. It hints that the azetidine ring is pH-responsive. Spectroscopic properties of **DAPA** and **DAP-Na** were also examined under the pH range of 4.0–8.6 (Appendix A). In Britton–Robinson buffer at pH~4.0, **DAPA** showed weak emission at 591 nm. When the pH increased from 6.0 to 8.0, the fluorescence intensity enhanced significantly along with a blue-shifted emission peak. It is anticipated that, with the increase of alkalinity, the acidic carboxyl group on **DAPA** was easily deprotonated and converted into the corresponding carboxylate. Based on the pH titration results, the p*Ka* value of **DAPA** was estimated to be 6.59. Concomitantly, the absorption maxima of **DAPA** showed a slight red shift above pH~8.0. **DAP-Na** also exhibited remarkable pH-dependent fluorescence emission properties. The fluorescence emission of **DAP-Na** was quenched dramatically as the pH decreased from 8.6 to 7.0 (Appendix A). Particularly, the fluorescence intensity was quite low when pH < 6.0, concomitantly with a red-shifted emission maximum. The pH investigations clearly demonstrate that these two single-benzene chromophores can be used as superiorly base- and acid-responsive fluorescent probes in aqueous media.

### 2.5. Sensitive and Discriminative Detection of Amino Acids

Next, to inspect the practical applications of these two highly pH-sensitive chromophores, the fluorescent sensing performances of **DAPA** and **DAP-Na** towards 20 natural amino acids with different p*I* values were examined (p*I* values of used amino acids: Arg, 10.76; Lys, 9.74; His, 7.59; Pro, 6.30; Ala, 6.02; Ile, 6.02; Leu, 5.98; Val, 5.97; Gly, 5.97; Trp, 5.89; Met, 5.75; Ser, 5.68; Tyr, 5.66; Gln, 5.65; The, 5.60; Phe, 5.48; Asn, 5.41; Cys, 5.02; Glu, 3.22; Asp, 2.97) [58,59]. As shown in Figure 4a, **DAPA** underwent a significant intensity increase along with ~30 nm blue-shifted emission in the presence of basic amino acids (100 μM, Arg and Lys). Addition of other natural amino acids induced negligible responses, although His induced a little emission enhancement in comparison with that of Arg or Lys. The fluorescence enhancement of **DAPA** upon addition of different amino acids revealed high selectivity to Arg and Lys (Figure 4b). Moreover, a visual fluorescence color change of the aqueous solution from dark red to bright yellow upon addition of Arg or Lys was clearly identified under UV lamp (*λ*_ex_ = 365 nm). As depicted in Appendix A, the selectivity of **DAP-Na** for Asp and Glu detection was also evaluated. When adding 20 natural amino acids to aqueous **DAP-Na**, only Asp and Glu produced dramatical fluorescence quenching as well as a 20 nm red shift. While, basic amino acids (Arg, Lys, and His) induced a slight emission enhancement of **DAP-Na** and all the other amino acids caused few changes. The fluorescence variation of **DAP-Na** to various amino acids also revealed the strong identifying capability to Asp and Glu (Appendix A).

Present single-benzene-based probes of **DAPA** and **DAP-Na** exhibited highly selective responses to basic (Arg and Lys) and acidic (Asp and Glu) amino acids, respectively. It is worthy to further appraise the anti-interference ability of the probes in the real-life applications. Therefore, a series of competitive experiments were conducted (Figure 4c,d and Appendix A). When Arg or Lys was added to the aqueous solution of **DAPA** with the co-existing interfering amino acids, 16 among 18 kinds of amino acids show no obvious effect. Asp and Glu show little interference because the released H^+^ might weaken the basicity of Arg or Lys, thus reducing the positive response of **DAPA** to Arg or Lys. The competitive experiments demonstrated that most of these natural amino acids did not significantly induce the fluorescent sensing of **DAPA** towards Arg or Lys. Similarly, we also verified the co-presence of other natural amino acids did not obviously influence the fluorescent detection of **DAP-Na** towards Asp or Glu (Appendix A).

Sensitivity is another crucial factor for determining the practical usability of the fluorescent probes. Accordingly, the sensing performances of **DAPA** for Arg and Lys were systematically studied. Figure 5a and Appendix A demonstrated that the fluorescence dependence of **DAPA** on Arg and Lys ranged from 2 to 200 μM, respectively. As depicted, with incremental addition of Arg or Lys to aqueous **DAPA** solution, the fluorescence intensity was gradually enhanced (turn-on) and accompanied by the emission peak shift from 591 to 558 nm (Δ*λ* = 33 nm). Appendix A represent the corresponding plots of the fluorescence intensity rations (I/I_0_) at 558 nm versus the concentrations of Arg and Lys. A relatively linear correlation was observed over a concentration range from 12 to 100 μM for both amino acids, and then the plot reached a plateau as the concentration of amino acids above 140 μM. To quantitatively evaluate the sensitivity of **DAPA**, the detection limits (DLs) were investigated using the standard IUPAC 3*δ* method [60]. The DL values for Arg and Lys were calculated to be 0.50 μM and 0.41 μM, respectively. In contrast, the fluorescence quenching (turn-off) responses of **DAP-Na** with respect to Asp and Glu were also carefully investigated (Figure 5b, Appendix A). Related DL values were found to be 0.12 μM for Asp and 0.16 μM for Glu, respectively. Compared with recently-reported fluorescence probes, the structures of **DAPA** and **DAP-Na** are the simplest but display efficient sensing performances to different amino acids in aqueous media (Appendix A).

Aiming to better understand the sensing mechanism, ^1^H NMR spectra were evaluated to study the interactions between the probes and the amino acids. As depicted in Figure 6, the signals of aromatic protons (H_a_) in **DAPA** were assigned to 7.46 ppm. After adding Arg, the aromatic protons shifted up-field obviously and the chemical shifts changed up to 0.33 ppm. Such results suggest **DAPA** was easily converted into the species of carboxylate anion through the deprotonation of basic Arg, thus increasing the electron cloud density on the benzene ring of **DAPA** [61]. Upon addition of Asp into the D_2_O solution of **DAP-Na**, all the protons assigned to **DAP-Na** shifted downfield (Appendix A). Specifically, the downfield shift magnitude of the aromatic proton signal (H_a′_) was close to 1.12 ppm, and that of methylene (H_b′_) attached to the nitrogen atom was close to 0.53 ppm. Such extraordinary shifts suggest that the Lewis-basic nitrogen of azetidine is most probably protonated by Asp, explaining why **DAP-Na** exhibited dramatically quenching phenomena toward acidic amino acids [45].

## 3. Materials and Methods

### 3.1. Reagents and Instruments

All the organic solvents for spectroscopic measurements were distilled prior to use. Water used was prepared from Milli-Q water. Diethyl 2,5-di(azetidine-1-yl)terephthalate was synthesized and characterized according to our previous report [44]. Twenty natural amino acids including arginine (Arg), lysine (Lys), histidine (His), aspartic acid (Asp), glutamic acid (Glu), proline (Pro), tryptophan (Trp), leucine (Leu), alanine (Ala), methionine (Met), glycine (Gly), cysteine (Cys), threonine (The), serine (Ser), tyrosine (Tyr), glutamine (Gln), asparagine (Asn), isoleucine (Ile), valine (Val), and phenylalanine (Phe) were purchased from J&K scientific company and used without further purification. Deuterated solvents of *d*_6_-DMSO and D_2_O were obtained from Cambridge Isotope Laboratories.

NMR spectra of the synthesized compounds were recorded on a Bruker 600 MHz spectrometer and their high-resolution mass spectra were determined by a Bruker maxis UHR-TOF mass spectrometer. UV-vis absorption spectra were measured on a Hitachi U-3900 spectrophotometer. Fluorescence measurements were performed on a time-correlated single-photon-counting FLS920 fluorescence spectrometer from Edinburgh Instruments. Absolute fluorescence quantum yields (Φ_f_) were measured on the Hamamatsu C9920-02G quantum efficiency measurer.

The femtosecond transient absorption (fs-TA) setup adopted in the present work was based on a PHAROS laser system from Light Conversion (1030 nm, ~200 fs, 200 μJ/pulse, and 100 kHz repetition rate), nonlinear frequency mixing techniques, and the Femto-TA100 spectrometer (Time-Tech Spectra) [62,63]. Briefly, the 1030 nm output pulse from the regenerative amplifier was split in two parts with an 80% beam splitter. The reflected part was used to pump an ORPHEUS Optical Parametric Amplifier (OPA) which generates a wavelength-tunable laser pulse from 300 nm to 15 μm. Here, 420 nm was used as the pump beam. The transmitted 1030 nm beam was split again into two parts. One part with less than 50% was attenuated with a neutral density filter and focused into a YAG window to generate a white light continuum from 500 to 1600 nm used for probe beam. The probe beam was focused with an Ag parabolic reflector onto the sample. After the sample, the probe beam was collimated and then focused into a fiber-coupled spectrometer with CMOS sensors and detected at a frequency of 10 kHz. The intensity of the pump pulse was controlled by a variable neutral-density filter wheel. The delay between the pump and probe pulses was controlled by a motorized delay stage. The pump pulses were chopped by a synchronized chopper at 5 kHz and the absorbance change was calculated with two adjacent probe pulses (pump-blocked and pump-unblocked).

Kinetic modeling: Kinetic modeling was carried out via target analysis on a composite data set of the fs-TA spectra in order to capture the complete dynamics. Using target analysis, the entire TA data set was fitted over the whole wavelength region and all the time delays with the application of a kinetic model. In this work, CarpetView (version 1.1.10) software was used for kinetic modelling of the transient absorption data (www.lightcon.com, accessed on 21 March 2022).

### 3.2. Synthesis of Compound ***DAPA***

Diethyl 2,5-di(azetidin-1-yl) terephthalate (0.1 g, 0.30 mmol) was dissolved in THF (1 mL) and methanol (3 mL), followed by addition of 2 M NaOH (aq. 2 mL). The reaction mixture was heated up to 85 °C and stirred for 6 h. Organic solvent was removed under reduced pressure, and the residue was neutralized with 1 M HCl solution until pH~3. Excess water was concentrated using a rotary evaporator at 40 °C and the residue was further purified by column chromatography (CH_2_Cl_2_/MeOH = 20/1, containing 0.5% AcOH) to afford **DAPA** (0.083 g, 79%). ^1^H NMR (600 MHz, *d*_6_-DMSO, ppm) *δ* 7.25 (s, 2H), 3.72 (t, *J* = 6 Hz, 4H), 3.25 (t, *J* = 6 Hz, 4H), 2.02 (m, 4H). ^13^C NMR (150 MHz, *d*_6_-DMSO, ppm) *δ* 169.26, 140.41, 117.45, 114.19, 43.08, 40.23, 31.52. ESI-HRMS *m*/*z*: [M + Cl]^−^, calc. for C_14_H_16_N_2_O_4_Cl: 311.0804, found: 311.0814.

### 3.3. Synthesis of Compound ***DAP-Na***

Diethyl 2,5-di(azetidin-1-yl) terephthalate (0.065 g, 0.20 mmol) was dissolved in THF (1 mL) and methanol (3 mL), followed by addition of 1 M NaOH (aq. 0.39 mL). The reaction mixture was heated up to 80 °C and stirred for 18 h. The reaction mixture was then cooled to room temperature and the precipitate was isolated by centrifugation, rinsed with CH_2_Cl_2_, and dried under reduced pressure to afford **DAP-Na** as a white solid (0.035 g, 56%). ^1^H NMR (600 MHz, D_2_O, ppm) *δ* 6.56 (s, 2H), 3.81 (t, 8H), 2.25 (m, 4H). ^13^C NMR (150 MHz, D_2_O, ppm) *δ* 177.51, 141.95, 127.87, 114.15, 54.09, 16.77. ESI-HRMS *m*/*z*: [M-Na]^−^, calc. for C_14_H_14_N_2_O_4_Na: 297.0857, found: 297.0865.

## 4. Conclusions

In summary, we developed two compact single-benzene-based chromophores, **DAPA** and its disodium carboxylate **DAP-Na**, which displayed favorable properties including good water solubility, large Stokes shift, intense luminescence in most polar and protonic solvents, and highly pH-responsive characteristics. Fs-TA spectra illustrated that quadrupolar **DAPA** usually involved the formation of an ICT state. The Frank–Condon state could be rapidly relaxed to a slight symmetry-breaking state upon light excitation following the solvent relaxation, then the slight charge separation may occur and the charge localization become partially asymmetrical in polar environment. DFT theoretical calculation results support with the experimental measurements well. Given these advantages, these two fluorescent probes were capable of rapid, highly selective, and sensitive detection of amino acids in aqueous media. **DAPA** can be successfully used in sensing Arg and Lys via fluorescence turn-on response and displayed DL values of 0.50 μM for Arg and 0.41 μM for Lys, respectively. In contrast, **DAP-Na** demonstrated a remarkable fluorescence quenching response toward Asp and Glu and the DLs were calculated to be 0.12 and 0.16 μM, respectively. The present work provides two structurally simple probes that help to understand their photophysic–structure relationship and render their fluorescent detection applications.

## Data Availability

Data is contained within the article or Appendix A.

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
