# Peer review of "Water-Soluble Single-Benzene Chromophores: Excited State Dynamics and Fluorescence Detection"

_molecules, 2022, doi:10.3390/molecules27175522_

Round 1

Reviewer 1 Report

In this manuscript, the authors the excited state dynamics and fluorescence detection properties of two single benzene chromophores. Overall the topic is interesting and the results are nice. However, no physical insight is presented in any point in the manuscript. The authors present the results as if they are writing a technical report and not a scientific article. Mainly, I would like to see more physical explanations on the observed properties. Below are some more specific comments.

1)      One of the molecules, exhibit significant changes in its absorption spectra by changing the solvent while the other one does not. I would expect some explanation on that.

2)      Most of my comments are related to the transient absorption spectroscopy methodology and results. There is not presentation of the experimental equipment used and of the analysis method. They present SADS spectra. So, is it a target model that has been used. What is the physical model they used? Why DMSO has been used as the solvent for the transient absorption measurements? Why not the other solvents? Why is the uncertainty regarding the lifetime given in the inset of figure 3b so large? The authors follow a well-known explanation for the interpretation of the TA spectra for quadrupolar molecules in which after excitation the excited state relaxes to a CT and symmetry broken Ct state aided by the solvent reaction field. However, this would lead to some spectral changes in the spectra within the first ps. However, the TA spectra exhibit no spectral changes but only a decay. So, to my opinion, the authors should reconsider their explanations. On the other hand, they attribute a time constant of 2543ps to solvent relaxation. However, solvent relaxation time are much smaller. The authors relate the long time constant found by TA spectroscopy with the lifetimes found by time-resolved fluorescence spectroscopy. But they do not show the time-resolved fluorescence measurements and their fittings. Overall, the TA spectroscopy part is very poor. And also, it is not related to the other part of the manuscript. It seems that it is a completely separate part of the manuscript.

3)      I would also expect a physical or a mechanistic explanation on why the fluorescence of DAPA is selective to Arg or Lys only. And I would also expect the same for DAP-Na.

4)      It is not clear what is shown in figure S8. It shows the results of which molecule? What is the difference between figure S8a,b and S9a,b? The paragraph of the manuscript describing these figures (lines 238-252) is not clear and should be written again.

In summary, the paper could be published after major revisions.

Author Response

Response to Reviewer #1

Comments:

In this manuscript, the authors the excited state dynamics and fluorescence detection properties of two single benzene chromophores. Overall, the topic is interesting and the results are nice. However, no physical insight is presented in any point in the manuscript. The authors present the results as if they are writing a technical report and not a scientific article. Mainly, I would like to see more physical explanations on the observed properties. Below are some more specific comments.

Response:

 Thanks a lot for your efforts to update our work. Proper updates have been made, hopefully our modification could satisfy your concerns. All the changes are marked in red for your convenience in the revised Manuscript and revised Supporting Information.

Major points:

Q-1. One of the molecules, exhibit significant changes in its absorption spectra by changing the solvent while the other one does not. I would expect some explanation on that.

A-1. Thanks for your good question. DAPA demonstrated two absorption bands located at 415-550 and 305-415 nm in both solvents of MeOH and EtOH. According to our understanding, the dicarboxylic acid of DAPA may be partially ionized in protic solvents, resulting in the presence of two components in the above solutions. While in water, the carboxy on one side is completely ionized, thus DAPA characterized single absorption maxima around 365 nm. However, for DAP-Na, there is only one component of carboxylate anion in the tested solvents. Related updates can be found in the revised Manuscript (c.f. Page 3).

Q-2. Most of my comments are related to the transient absorption spectroscopy methodology and results. There is not presentation of the experimental equipment used and of the analysis method. They present SADS spectra. So, is it a target model that has been used. What is the physical model they used? Why DMSO has been used as the solvent for the transient absorption measurements? Why not the other solvents?

A-2. Thanks for your advice. The setup of femtosecond transient absorption (fs-TA) spectroscopy has been added in the section 3.1 (c.f. Page 9) and marked in red. Kinetic modeling was carried out via target analysis on a composite data set of the fs-TA and ns-TA spectra in order to capture the complete dynamics. Using target analysis, the entire TA data set is fitted over all the wavelengths and all the time delays with the application of a kinetic model. CarpetView (version 1.1.10) is a software package for viewing, processing and analyzing the data from ultrafast spectroscopic experiments (www.lightcon.com).

For another question, DAPA is very soluble in DMSO and its absorption spectra indicated that only one component of dicarboxylic acid state is presented in the solution. On the other hand, the use of DMSO as solvent could effectively avoid some specific interactions between the fluorophore and the solvent. Once again, when use DMSO as solvent, DAPA and its reference fluorophore can be efficiently excited with fs-420 nm laser.

Q-3. Why is the uncertainty regarding the lifetime given in the inset of figure 3b so large? The authors follow a well-known explanation for the interpretation of the TA spectra for quadrupolar molecules in which after excitation the excited state relaxes to a CT and symmetry broken CT state aided by the solvent reaction field. However, this would lead to some spectral changes in the spectra within the first ps. However, the TA spectra exhibit no spectral changes but only a decay. So, to my opinion, the authors should reconsider their explanations. On the other hand, they attribute a time constant of 2543ps to solvent relaxation. However, solvent relaxation time are much smaller. The authors relate the long time constant found by TA spectroscopy with the lifetimes found by time-resolved fluorescence spectroscopy. But they do not show the time-resolved fluorescence measurements and their fittings. Overall, the TA spectroscopy part is very poor. And also, it is not related to the other part of the manuscript. It seems that it is a completely separate part of the manuscript.

A-3. Thanks for your good question. We have rewritten the discussion of fs-TA results. Briefly, the Frank-Condon S1 state could be rapidly relaxed to a slight symmetry-broking state upon light excitation following the solvent relaxation and structure relaxation, then the slight charge separation may occur and the charge localization became partially asymmetrical in polar environment. The global fitting of the DAPA data collected in DMSO illustrated the observation of three decay processes (Figures 3b, 3d, S3a-3b). The first decay lifetime may reflect the time-constant of structure relaxation and solvent relaxation (τ0-1 ~ 4.1±0.1 ps). The second one is compatible with the time-constant of excited-state induced symmetry-breaking (τ1-2 ~ 2543±800 ps), implying the ICT localizing on the partial donor (azetdiine)-acceptor (carboxy) branch. The third decay of the fully relaxed singlet state via fluorescence emission decay lasted around 7.0 ns, which is consistent with the time-resolved fluorescence measurement result of DAPA in DMSO (7.65 ns in DMSO, Table 1). Related updates can be found in the revised manuscript (c.f. Page 5).

Q-4. I would also expect a physical or a mechanistic explanation on why the fluorescence of DAPA is selective to Arg or Lys only. And I would also expect the same for DAP-Na.

A-4. Thanks for your good question. Based on our understanding, the underlying sensing mechanism should be the Lewis-basic nitrogen of azetidine most probably protonated by Asp, explaining why DAP-Na exhibited dramatically quenching phenomena toward acidic amino acids, as discussed in last part of section 2.5. In details, as shown in Figure 6, 1H NMR spectra were performed to study the interactions between the probes and the amino acids. The signals of aromatic protons (Ha) in DAPA were assigned to 7.46 ppm. After adding Arg, the aromatic protons shifted up-field obviously and the chemical shifts changed up to 0.33 ppm. Such results suggest DAPA was easily converted into the species of carboxylate anion through the deprotonation of basic Arg, thus increasing the electron cloud density on the benzene ring of DAPA. Upon addition of Asp into the D2O solution of DAP-Na, all the protons assigned to DAP-Na shifted downfield (Figure S9). Specifically, the downfield shift magnitude of the aromatic proton signal (Ha') was close to 1.12 ppm, and that of methylene (Hb') attached to the nitrogen atom was close to 0.53 ppm.

Q-5. It is not clear what is shown in Figure S8. It shows the results of which molecule? What is the difference between figure S8a, b and S9a, b? The paragraph of the manuscript describing these figures (lines 238-252) is not clear and should be written again.

A-5. Thanks a lot. The original Figure S8 and S9 depicted the same meanings with different y-axes, one is the intensity and the other is the ration of I/I0. We removed the original Figure S9 in the first version and also updated the expressing sentences in the revised Manuscript (c.f. Page 7).

Reviewer 2 Report

In the manuscript "Water-Soluble Single-Benzene Chromophores: Excited State Dynamics and Fluorescence Detections" the authors propose two fluorescent dyes as probes for detection of arginine, lysine and glutamic acid. The proposed dyes are new compounds derived from a similar compound with the same fluorofore. The study on the fluorescent properties of the parent dye has been recently published by the authors (Ref. 44 in the manuscript). The novelty of this work is thus in sensing application. So my major concern is in the usability of the proposed dyes as fluorescence-based sensors. The authors attribute the observed changes in the fluorescence of the dyes upon addition of the analytes to pH dependence of the fluorescence response of the dye, which sounds reasonable. What is the dependance of the detection limits for Arg, Lys, Asp and Glu on the pH? Can the pH of the real sample affect the fluorescence response and distort the concentration measurements? Can the presence of amines or other bases affect the results of the measurements?

My other question concerns the transcient spectroscopy experiments. Please explain how "Fs-TA spectra illustrated quadrupolar DAPA usually involved the formation of an ICT state" (Section 4, Conclusions). The global fitting of the transient spectroscopy data gives information on three components in the evolution of the response, but do not give information on what these components are. Also, according to Fig. 3d, the three-exponents model seems to be not a perfect match to the experimental data. Did you try other models to fit the transient spectroscopy data?

In Section 4, Conclusions, the authors state that "the charge separation localized on one side of the excited chromophores." Please explain this statement. Probably, this is not the best wording.

Minor remarks:

- Please place the Figures after the text they illustrate, not before.

- Please add the description of the time-resolved spectroscopy setup to Section 3.1.

- According to the NMR spectra, the symmetry of DAPA is lower than the symmetry of DAP-Na. Probably, this should be commented in the maintext.

- Section 3.3.: according to Fig. S13, the signal at 2.25 is not a triplet.

- There are a few typos in the manuscript, please check it.

Author Response

Response to Reviewer #2

Comments:

In the manuscript "Water-Soluble Single-Benzene Chromophores: Excited State Dynamics and Fluorescence Detections", the authors propose two fluorescent dyes as probes for detection of arginine, lysine and glutamic acid. The proposed dyes are new compounds derived from a similar compound with the same fluorophore. The study on the fluorescent properties of the parent dye has been recently published by the authors (Ref. 44 in the manuscript). The novelty of this work is thus in sensing application. So my major concern is in the usability of the proposed dyes as fluorescence-based sensors. The authors attribute the observed changes in the fluorescence of the dyes upon addition of the analytes to pH dependence of the fluorescence response of the dye, which sounds reasonable.

Response:

Thanks a lot for your efforts to update our work. Proper updates have been made, hopefully our modification could satisfy your concerns. All the changes are marked in red for your convenience in the revised Manuscript and revised Supplementary material.

Remarks:

Q-1. What is the dependance of the detection limits for Arg, Lys, Asp and Glu on the pH? Can the pH of the real sample affect the fluorescence response and distort the concentration measurements? Can the presence of amines or other bases affect the results of the measurements?

A-1. Thanks for your good question. We agree with you that the pH and some acids and amines should have effect on the sensing results. Because the underlying sensing mechanism should be the Lewis-basic nitrogen of azetidine most probably protonated by Asp, explaining why DAP-Na exhibited dramatically quenching phenomena toward acidic amino acids, as discussed in last part of section 2.5. In details, as shown in Figure 6, 1H NMR spectra were performed to study the interactions between the probes and the amino acids. The signals of aromatic protons (Ha) in DAPA were assigned to 7.46 ppm. After adding Arg, the aromatic protons shifted up-field obviously and the chemical shifts changed up to 0.33 ppm. Such results suggest DAPA was easily converted into the species of carboxylate anion through the deprotonation of basic Arg, thus increasing the electron cloud density on the benzene ring of DAPA. Upon addition of Asp into the D2O solution of DAP-Na, all the protons assigned to DAP-Na shifted downfield (Figure S9). Specifically, the downfield shift magnitude of the aromatic proton signal (Ha') was close to 1.12 ppm, and that of methylene (Hb') attached to the nitrogen atom was close to 0.53 ppm.

 To be honest, in section 2.4, we conducted the pH effect ranged from 4.0 to 8.6 on the sensing results. The pH investigations clearly demonstrate that these two single-benzene chromophores can be used as superiorly base- and acid-responsive fluorescent probe in aqueous media in the special pH conditions.

Q-2. My other question concerns the transient spectroscopy experiments. Please explain how "Fs-TA spectra illustrated quadrupolar DAPA usually involved the formation of an ICT state" (Section 4, Conclusions). The global fitting of the transient spectroscopy data gives information on three components in the evolution of the response, but do not give information on what these components are. Also, according to Fig. 3d, the three-exponents model seems to be not a perfect match to the experimental data. Did you try other models to fit the transient spectroscopy data?

A-2. Thanks for your good question. The global fitting of the fs-TA data collected in DMSO illustrated the observation of three decay processes, not three components. We have updated the expression in the revised Manuscript (c.f. Page 5). In detail, the Frank-Condon S1 state could be rapidly relaxed to a slight symmetry-breaking state upon light excitation following the solvent relaxation and structure relaxation, then the slight charge separation may occur and the charge localization became partially asymmetrical in polar environment. The global fitting of the DAPA data collected in DMSO illustrated the observation of three decay processes (Figures 3b, 3d, S3a-3b). The first decay lifetime may reflect the time-constant of structure relaxation and solvent relaxation (τ0-1 ~ 4.1±0.1 ps). The second one is compatible with the time-constant of excited-state induced symmetry-breaking (τ1-2 ~ 2543±800 ps), implying the ICT localizing on the partial donor (azetdiine)-acceptor (carboxy) branch. The third decay of the fully relaxed singlet state via fluorescence emission decay lasted around 7.0 ns, which is consistent with the time-resolved fluorescence measurement result of DAPA in DMSO (7.65 ns in DMSO, Table 1). Related updates can be found in the revised manuscript (c.f. Page 5)

Q-3. In Section 4, Conclusions, the authors state that "the charge separation localized on one side of the excited chromophores." Please explain this statement. Probably, this is not the best wording.

A-3. Thanks for your reminder. We have updated it to be “Its Frank-Condon state (S1) rapidly relaxed to a symmetry-broken state upon light excitation and solvent relaxation, thus the resultant charge separation may occur and the charge localization became partially asymmetrical.” Related changes marked in red can be found in the Pages 5 and 10 in the revised Manuscript.

Q-4. Please place the Figures after the text they illustrate, not before.

A-4. Thanks a lot. We have rearranged all the figures.

Q-5. Please add the description of the time-resolved spectroscopy setup to Section 3.1.

A-5. The details of the femtosecond transition absorption to characterize the time-resolved spectroscopies have been added in the revised Manuscript (c.f. Page 9). Meanwhile, two related references numbered #58 and #59 were provided properly.

Q-6. According to the NMR spectra, the symmetry of DAPA is lower than the symmetry of DAP-Na. Probably, this should be commented in the maintext.

A-6. Thanks for your good question. After a first glance, both chromophores of DAPA and DAP-Na are symmetrical, so the proton close to the N atom in the azetidine ring specially in DAPA should not be split as DAP-Na (Figure S12). While as shown in Figure S10 for DAPA, there are two groups of split peaks at 3.72 ppm and 3.25 ppm assigned to the proton (-NCH2-) close to the N atom. In our opinion, this special splitting should be originated from the formed hydrogen bonding between the carboxyl-H and nitrogen, the generated six-member-like ring induced the asymmetry or the twist of the azetidine ring. Therefore, two groups of split peaks at 3.72 ppm and 3.25 ppm assigned to the proton (-NCH2-) close to the N atom were obtained. Related discussion was added in the revised Manuscript (c.f. Page 5). It also hints that the azetidine ring is pH-responsive.

Q-7. Section 3.3.: according to Fig. S13, the signal at 2.25 is not a triplet.

A-7. Thanks for pointing out the mistake. We corrected it to be multiple peaks (c.f. page 9 in the revised Manuscript).

Q-8. There are a few typos in the manuscript, please check it.

A-8. Thanks again. As suggested, we have checked the whole writing carefully and some corrections are marked in red in the revised Manuscript.

Round 2

Reviewer 1 Report

In the revised manuscript, the authors tried to address the comments in order to strengthen their work. However, I still have strong concerns about the transient absortion measurements and conclusions. The authors conclude based on the fs-TA spectra that the FC state rapidly relaxes to a SB state upon solvent relaxation. But this is not supported by the spectra which do not show any relaxation. Specifically, they only show decay to the ground state without any spectral changes which could be due to excited state relaxation. To me any relaxation phenomena may be ultrafast and may not be captured. Besides, the authors attribute the 2.54 ns time constant to SB which however is not supported by the literature since SB typically occurs in a few ps. Overall, it is not straightforward to make any conclusions taking into account the fs-TA spectra as presented here. I would strongly suggest to make more measurements in at least two more solvents with diferent polarity and or proticity.  

Author Response

Response: Thanks a lot for your good advice to update this work. As suggested, femtosecond transient absorption (fs-TA) measurements of DAPA were further carried out in three other polar solvents including DMF, EtOH, and MeOH upon an excitation at fs-420 nm. In brief, to provide an overview of excited-state dynamics and charge transfer properties of the fluorophores, the decay curves in fs-TA spectra and multi-exponential fitting kinetic traces probed at different wavelengths were compared in Figure 3b and Figures S3-S5. The corresponding lifetimes of transient states were summarized in Table S1. Based on these spectral traces and time constants, a corresponding sequential model for global fitting was proposed as shown in Figure 3d. The first decay lifetime may reflect the time-constant of solvent relaxation (tSR<1 ps).  The second one is associated with the formation of the charge separation state and symmetry-breaking state (tCS~5 ps), implying the ICT localizing on the partial donor (azetdiine)-p-acceptor (carboxy) branch. The falling processes reflected the formation of the charge recombination process and the quenching of the excited state absorption (ESA) process (Table S1). Related updates highlighted in red can be found in the revised Manuscript (c.f. Page 5) and the revised Supporting Information (Figure S3-S5).

p. s. Supplementary transient absorption (TA) spectra and data

Reviewer 2 Report

In the revised version the authors did some amendation. Yet, my main concern regarding the applicability of the studied fluorescence dyes as amino acid sensitive sensors remained unresolved. In the reply to Q1, the authors state that "The pH investigations clearly demonstrate that these two single-benzene chromophores can be used as superiorly base- and acid-responsive fluorescent probe in aqueous media in the special pH conditions." However, this only confirms that the fluorescence response is pH sensitive and gives no information on how to separate the pH-caused response from the response which is due to the presence of the amino acid.

The interpretation of the observed solvochromic effect needs revision. In lines 115-118 (p. 3), the authors explain the observed change in the absorption spectra of DAPA by "partial ionization" of DAPA in protic solvents and "complete ionization" in  water. First of all, water is a protic solvent. Also, DAP-Na is a salt, and the bond O-Na is more ionic in nature than O-H bond in DAPA, while for DAP-Na no solvatochromic effect was observed by the authors. What do the authors mean by "partial ionized" and "completely ionized" (lines 116 and 118, p. 3)?

Minor remark: I guess, Fig. 1a depicts excitation and emission spectra, not absorption and  emission spectra. Please check if that is correct.

Author Response

Reviewer #2

Q-1. In the revised version the authors did some amendation. Yet, my main concern regarding the applicability of the studied fluorescence dyes as amino acid sensitive sensors remained unresolved. In the reply to Q1, the authors state that "The pH investigations clearly demonstrate that these two single-benzene chromophores can be used as superiorly base- and acid-responsive fluorescent probe in aqueous media in the special pH conditions." However, this only confirms that the fluorescence response is pH sensitive and gives no information on how to separate the pH-caused response from the response which is due to the presence of the amino acid.

A-1. The pH-dependent fluorescent measurements demonstrated that when pH increased from 6.0 to 8.0, the fluorescence intensity of DAPA enhanced significantly along with a blue-shifted emission peak. Considering arginine (Arg) and Lysine (Lys) are two alkaline amino acids among the 20 common amino acids with pI values of 10.76 and 9.74, respectively (pI values of other used amino acids: His, 7.59; Thr, 6.53; Pro, 6.30; Ala, 6.02; Ile, 6.02; Leu, 5.98; Val, 5.97; Gly, 5.97; Trp, 5.89; Met, 5.75; Ser, 5.68; Tyr, 5.66; Cys, 5.02; Glu, 3.22; Asp, 2.97) (c.f. Dyes Pigm. 2020, 175, 108131; Sens. Actuators B, 2017, 241, 1270-1275; Chem. Eur. J. 2014, 20, 4661-4670.), these two alkaline amino acids could be selectively detected over other amino acids by using DAPA as a fluorescent probe when the pH is higher than 7.0. Furthermore, the fluorescence emission of DAP-Na is quenched dramatically as pH decreases from 8.6 to 7.0, which could be applied for recognition of acidic amino acids of Asp and Glu from other common amino acids in aqueous solution. Related updates can be found in the revised Manuscript (c.f. Page 7).

Q-2. The interpretation of the observed solvochromic effect needs revision. In lines 115-118 (p. 3), the authors explain the observed change in the absorption spectra of DAPA by "partial ionization" of DAPA in protic solvents and "complete ionization" in water. First of all, water is a protic solvent. Also, DAP-Na is a salt, and the bond O-Na is more ionic in nature than O-H bond in DAPA, while for DAP-Na no solvatochromic effect was observed by the authors. What do the authors mean by "partial ionized" and "completely ionized" (lines 116 and 118, p. 3)?

A-2. Thanks for your good question. DAPA demonstrated two absorption bands located at 415-550 and 305-415 nm in both solvents of MeOH and EtOH. While it showed single absorption maxima around 457 nm in DMF and 470 nm in DMSO, respectively. In contrast, DAPA in water characterized single absorption maxima around 365 nm (c.f. Figure 1b in Page 3). The different UV-vis absorption performances obviously showed the solvent-dependent properties of DAPA. Related to the underlying structure-property relationship, we think that the dicarboxylic acid of DAPA may be partially ionized in EtOH and MeOH, resulting in the presence of two components (ionized state and carboxylic acid state) in the above solutions. If considering the polarity of MeOH is larger than that of EtOH, the proportion of the component of ionized state (377 nm) is significantly higher than that of carboxylic acid state (456 nm) as shown in Figure 1b. While in water, the dicarboxylic acid of DAPA may be completely ionized, thus DAPA characterized single absorption maxima around 365 nm. However, for DAP-Na, whether it is ionized or not in the tested solvents, its electronic structure or molecular orbital distributions of HOMO and LUMO does not change. Thus, no solvatochromic effect was observed in the UV-vis absorption spectra of DAP-Na in various solvents (Figure S2a). Related updates can be found in the revised Manuscript (c.f. Page 3).

Q-3. Minor remark: I guess, Fig. 1a depicts excitation and emission spectra, not absorption and emission spectra. Please check if that is correct.

A-3. Thanks a lot. We double checked Fig. 1 once again. Fig. 1a indeed depicted the normalized absorption and emission spectra of DAPA and DAP-Na in aqueous solution. While Fig. S1a in the Supporting Information showed their normalized fluorescence excitation and emission spectra in water.
